# Acute Respiratory Distress Syndrome Definitions in Adults and Children: A Comparative Narrative Review

**DOI:** 10.3390/jcm14217644

**Published:** 2025-10-28

**Authors:** Patricio Gonzalez-Pizarro, Fernando Suarez-Sipmann

**Affiliations:** 1Pediatric Anesthesia and Critical Care Department, La Paz University Hospital, 28046 Madrid, Spain; 2IdiPaz Research Institute, 28046 Madrid, Spain; 3Intensive Care Medicine Department, La Princesa University Hospital, 28006 Madrid, Spain; 4Centro de Investigación Biomédica en Red de Enfermedades Respiratorias (CIBERES), Instituto de Salud Carlos III, 28029 Madrid, Spain; 5Fundación Para la Investigación Biomédica, Hospital Universitario de La Princesa, 28006 Madrid, Spain

**Keywords:** ARDS, PARDS, definitions, diagnostic criteria, consensus, review

## Abstract

**Background**: Acute Respiratory Distress Syndrome (ARDS) was first described in 1967 by Ashbaugh et al. as a severe acute hypoxemic respiratory failure with reduced lung compliance, representing a common end-path of severe pulmonary endothelial inflammation from diverse etiologies. Since then, several definitions for the adult syndrome have been proposed, culminating in the 2024 “New Global Definition” (Berlin 2.0). In pediatrics, dedicated criteria (pediatric ARDS, PARDS) have been established over the past decade, with the most recent update published by the Second Pediatric Acute Lung Injury Consensus Conference (PALICC-2) in 2023. **Methods**: We performed a narrative literature review of consensus statements and key studies defining ARDS in adult and pediatric (non-neonatal) populations. Primary sources included the full Berlin 2.0 and PALICC-2 documents, supplemented by PubMed, Embase, and society guidelines. Definitions were compared across major diagnostic domains: timing of onset, imaging requirements, oxygenation thresholds, inclusion of patients with chronic comorbidities, ventilatory support modalities, and applicability in resource-limited settings. **Results**: Both definitions show convergence in incorporating non-invasive oxygenation indices and adaptability to resource-limited contexts. Key distinctions include the use of the Oxygenation Index (OI) or Oxygen Saturation Index (OSI) in invasively ventilated pediatric patients—metrics that integrate mean airway pressure and correlate more strongly than PaO_2_/FIO_2_ with short-term outcomes—and PALICC-2’s explicit inclusion of patients with chronic lung disease or cyanotic congenital heart disease when acute deterioration is documented. Imaging criteria differ, with Berlin 2.0 requiring bilateral opacities (and permitting lung ultrasound) versus PALICC-2’s acceptance of unilateral findings. **Conclusions**: Berlin 2.0 and PALICC-2 represent substantial progress toward globally applicable ARDS definitions, but physiologic and structural differences remain. These distinctions have prognostic and research implications, and harmonization will be critical to improve cross-age comparability, optimize clinical trial design, and ultimately enhance patient outcomes.

## 1. Introduction

Acute Respiratory Distress Syndrome (ARDS) is a life-threatening condition characterized by acute severe hypoxemic respiratory failure secondary to non-cardiogenic pulmonary edema, first described in 1967 by Ashbaugh et al. [1]. Since then, its clinical definition has undergone multiple revisions to improve diagnostic accuracy and standardization for both clinical care and research (Figure 1). The most widely adopted adult definition, the Berlin definition, was established in 2012 [2,3], but its limitations [4]—particularly in resource-limited settings lacking chest radiography, arterial blood gas analysis, or mechanical ventilation—led to adaptations such as the Kigali modification [5].

Recently, the wider use of non-invasive oxygenation indices [9,10,11,12,13], High-Flow Nasal Oxygen (HFNO), especially during COVID-19 [14,15,16], and lung ultrasound have challenged the Berlin criteria and prompted calls to include these modalities in ARDS definitions [17,18,19,20,21]. To address these shifts in practice, a global consensus conference in 2021 produced the 2024 “New Global Definition” (Berlin 2.0) [6], expanding diagnostic inclusivity while maintaining comparability with previous definitions.

Pediatric ARDS (PARDS) presents additional challenges due to age-specific physiology, epidemiology, and management considerations. The first pediatric-specific definition, proposed by the Pediatric Acute Lung Injury Consensus Conference (PALICC) Group in 2015 [7,22,23], was followed by an international description of PARDS incidence and epidemiology [24]. Subsequent years saw advances in pathobiology, lung-protective strategies, and supportive technologies [25,26,27], alongside variability in adoption and implementation across settings, with implications for outcomes [28,29,30]. The updated PALICC-2 criteria [8] incorporate physiological refinements, explicit guidance for patients with chronic comorbidities, and adaptations for resource-limited environments, while also integrating advances in digital monitoring, automated data capture, and multicenter data-sharing platforms.

Despite this progress, important discrepancies remain between adult and pediatric ARDS definitions—most notably in oxygenation thresholds, imaging criteria, and the scope of patients they encompass. Understanding these differences is essential not only for accurate diagnosis and prognostication, but also for harmonizing research, facilitating cross-age comparisons, helping improving patient outcomes. This manuscript compares Berlin 2.0 and PALICC-2 across key diagnostic domains, evaluating their strengths, weaknesses, and implications for future research and clinical practice.

## 2. Materials and Methods

We conducted a literature review to identify and compare the most recent consensus definitions of ARDS in adults and pediatric patients. Searches were performed in MEDLINE/PubMed, Embase, Web of Science Core Collection, Scopus, and the Cochrane Library for records from 1 January 1967 to 31 August 2025. Full search strings and key words are listed in Appendix A. We included peer-reviewed guidelines, consensus statements, systematic reviews, and original studies that proposed, evaluated, or compared ARDS definitions (adult or pediatric). We excluded non-human studies, non-peer-reviewed sources, single-patient case reports without discussion of diagnostic criteria, and non-English/Spanish articles.

The 2024 “New Global Definition” (Berlin 2.0) for adult ARDS [6] and the 2023 Second Pediatric Acute Lung Injury Consensus Conference definition [8] were selected as the primary reference frameworks. Full-text documents for both definitions were reviewed in detail, focusing on their diagnostic domains, inclusion and exclusion criteria, and adaptations for resource-limited settings.

Comparative analysis was structured around oxygenation metrics, imaging features, cardiac criteria, patient population inclusivity, and diagnostic pathways according to type of respiratory support. Additional literature was reviewed to contextualize prognostic relevance, particularly of the oxygenation index (OI) in pediatric populations, and to identify strengths, weaknesses, and potential areas for harmonization. Findings were synthesized into summary tables and narrative sections to facilitate interpretation for both clinical and research applications.

## 3. Results

Key similarities and differences between the two aforementioned definitions emerged across multiple diagnostic domains, as detailed below.


**Oxygenation criteria**


In adults, the primary metric for defining hypoxemia is the arterial oxygen tension to inspired oxygen fraction ratio (PaO_2_/FIO_2_, in mmHg). Severity categories for intubated patients are: mild (>200–≤300), moderate (≤200–>100), and severe (≤100). When arterial blood gases are unavailable, the SpO_2_/FIO_2_ ratio can be used, with corresponding thresholds: mild (>235–≤315), moderate (≤235–>148), and severe (≤148). Patients on Non-Invasive Mechanical Ventilation (NIV), CPAP or HFNO therapy (≥30 L/min) qualify for ARDS criteria when the PaO_2_/FIO_2_ < 300 mmHg or SpO_2_/FIO_2_ < 315 (if SpO_2_ < 97%). The latest update includes an additional category for resource-limited settings where neither arterial blood gas, HFNO, NIV, and/or mechanical ventilation are routinely available. In this case, an SpO_2_/FIO_2_ ≤ 315 is indicative to trigger an ARDS ventilatory approach.

In pediatric patients, PALICC-2 uses the Oxygenation Index (OI) (OI = mean airway pressure (MAP) (cmH_2_O) × FIO_2_/PaO_2_ (mmHg)) or the OSI (OSI = MAP × FIO_2_/SpO_2_) as the preferred metrics when invasive ventilation is in place. PARDS is defined when the OI ≥ 4 or OSI ≥ 5, with only two severity groups: mild/moderate: OI < 16 or OSI <12, and severe: OI ≥ 16 or OSI ≥ 12. Patients on NIV support are categorized as follows: mild/moderate: PaO_2_/FIO_2_ > 100 or SpO_2_/FIO_2_ > 150; severe: PaO_2_/FIO_2_, ≤100 or SpO_2_/FIO_2_ ≤ 150. Additionally, PALICC-2 includes two categories for patients with “Possible PARDS” (Nasal CPAP/NIV or HFNO (≥1.5 L/kg/min or ≥30 L/min) with PaO_2_/FIO_2_, ≤300 or SpO_2_/FIO_2_ ≤ 250) and “At risk for PARDS” (Any interface with and oxygen supplementation to maintain SpO_2_ ≥ 88% but not meeting definition for PARDS or possible PARDS) provided that the respiratory failure is not justified solely from airway obstruction (e.g., asthma, bronchospasm). These differences are summarized in Table 1.


**Imaging criteria**


Berlin 2.0 requires the presence of bilateral opacities on chest radiograph, Computed Tomography (CT), or lung ultrasound (LUS), not fully explained by effusions, collapse, or nodules. PALICC-2, in contrast, accepts unilateral or bilateral opacities, recognizing that focal disease patterns are more common in children. Imaging modalities in PALICC-2 are limited to radiography or CT, whereas lung ultrasound is not included. The inclusion of LUS in adult criteria may expand diagnostic access in resource-limited settings but introduces potential operator-dependent variability.


**Hemodynamic criteria**


In adults, ARDS diagnosis requires that respiratory failure is not fully explained by cardiac failure or fluid overload, and objective assessment (e.g., echocardiography) should be used when available to exclude the predominance of a hydrostatic edema. In pediatrics, left ventricular dysfunction or cyanotic congenital heart disease do not preclude a PARDS diagnosis, as long as there is a known precipitating acute pulmonary insult and a documented functional deterioration from baseline oxygenation status. Both definitions acknowledge the frequent overlapping of respiratory and hemodynamic pathophysiologic features and propose more rational and clinically adequate criteria than past definitions.


**Other diagnostic considerations**


Both definitions require onset within one week of a known clinical insult or new/worsening respiratory symptoms, and both allow clinical judgment when precise timing is unclear. Both accept a range of underlying etiologies, but PALICC-2 is explicit in including chronic pulmonary or cardiac conditions when criteria for acute deterioration are met. Additionally, as previously commented, both definitions now incorporate considerations for resource-limited settings, adjusting oxygenation thresholds when advanced monitoring is not available.

Neither the Berlin nor the PALICC definitions incorporate sex-specific criteria. However, observational studies have identified gender-related disparities in ARDS management [31]. Women, especially those of shorter stature, are less likely to receive low-tidal-volume ventilation because predicted body weight is derived from height and sex, and errors in height estimation disproportionately affect them [32,33]. Although ARDS incidence appears higher in men and adjusted mortality is generally similar, some analyses suggest worse outcomes among women with severe ARDS [31]. Pregnancy introduces additional physiologic considerations [34], but pregnant patients remain systematically excluded from major ARDS trials, leaving an important evidence gap.

## 4. Discussion

The updated Berlin 2.0 and PALICC-2 definitions represent the most current consensus frameworks for diagnosing ARDS in adults and children, respectively, reflecting substantial advances in physiologic understanding, monitoring capabilities, and adaptability to diverse clinical environments. While both share common structural elements—such as onset criteria, inclusivity of multiple etiologies, and adaptations for low-resource settings—they diverge in key domains that have direct implications for prognosis, research comparability, and bedside applicability (Figure 2). Central to this divergence is the choice of oxygenation metrics, with Berlin 2.0 retaining the PaO_2_/FIO_2_, ratio as its primary tool, and PALICC-2 prioritizing the Oxygenation Index for invasively ventilated patients. The following discussion examines some of the limitations of current criteria, the prognostic role of OI in pediatric populations comparing the operational strengths and limitations of each definition.


**Oxygenation criteria in the ARDS definition**


As mentioned, oxygenation criteria represented by the PaO_2_/FIO_2_, as a categorization of the deterioration in lung function, remain the cornerstone of the ARDS definition in adults. In favor is the abundance of data coming from the numerous clinical trials using this criterion. There are however several unsolved shortcomings. The relationship between PaO_2_ and FIO_2_ is non-linear, especially at lower levels of FIO_2_, limiting its validity when assessed in non-standardized conditions as is the case in the Berlin definition. PaO_2_/FIO_2_, can vary greatly depending on the FIO_2_ used as most of the lung regions in ARDS function as low ventilation to perfusion ratio units. The ratio also changes in response to time spent on mechanical ventilation and ventilator settings, particularly PEEP. Although a minimum PEEP of 5 cmH_2_O is mandated in the Berlin definition, the clinical heterogeneity in ventilatory strategies and PEEP titration limits the reliability of this criterion. Furthermore, the use of SpO_2_ instead of PaO_2_ increases this shortcoming with a potential larger impact in resource limited environments. Therefore, future revisions of oxygenation criteria, especially in the adult population, should consider a standardized assessment, for instance as the one proposed by Villar et al. [35] who demonstrated that the assessment of ARDS criteria combining a PEEP of 10 cmH_2_O and an FIO_2_ ≥ 0.5, 24 after initiation of mechanical ventilation was more closely related to ARDS severity than non-standardized assessments.


**Relevance of Oxygenation Index (OI) in Mechanically Ventilated Pediatric Patients**


Multiple studies have demonstrated that serial measurements of OI during the early course of PARDS provide superior prognostic accuracy compared to a single baseline assessment [23,36,37]. In particular, OI measured at 24–72 h after diagnosis has been shown to refine mortality risk stratification, with the 24-h value emerging as a simple and highly accurate predictor of 30-day survival [38,39]. In pediatric oncology patients with PARDS, for example, each incremental increase in OI at 24 h was independently associated with a higher hazard of death at 30 days, with pragmatic thresholds of OI ≥ 10 or PaO_2_/FIO_2_ ≤ 150 identifying mortality rates of approximately 50%, compared to ~10% in patients below these cutoffs [40]. Similar trends have been reported in general PICU cohorts, where OI correlates more strongly than PaO_2_/FIO_2_ with illness severity, multi-organ dysfunction, and the need for advanced support such as ECMO [41,42].

The non-invasive surrogate OSI, which substitutes SpO_2_ for PaO_2_, has been shown to reproduce much of OI’s prognostic capability when arterial blood gases are unavailable [43,44]. OSI-based severity categories align well with OI and can be monitored continuously at the bedside, though they inherit the limitations of pulse oximetry—including reduced accuracy at high saturations, in low perfusion states, or in patients with darker skin pigmentation [45,46,47].

Beyond short-term mortality, OI trajectories during the first week of illness have been associated with other clinically relevant outcomes, including ventilator-free days and the resolution of PARDS [48,49]. A persistently elevated or rising OI despite optimized ventilatory management signals refractory disease and is often used to prompt evaluation for rescue therapies such as ECMO. While there is no universally accepted OI threshold for initiating ECMO in pediatrics, many centers consider referral when OI remains high despite optimal lung-protective strategies, with decisions guided by the broader clinical context [50,51].

Evidence linking OI directly to long-term functional or neurocognitive outcomes is less robust. Survivors of PARDS, particularly those requiring prolonged mechanical ventilation, may experience deficits in physical function, health-related quality of life, and neurodevelopment, but these sequelae appear to result from a complex interplay of factors—including the severity and duration of hypoxemia, sedation exposure, delirium, immobility, and systemic inflammation—rather than OI alone [52,53,54]. Current data are insufficient to establish specific OI thresholds that predict neurologic injury, highlighting an important gap for future research.

Therefore, OI—and its non-invasive counterpart OSI—provides a more granular and prognostically relevant measure of respiratory failure severity in mechanically ventilated children than PaO_2_/FIO_2_ alone (Figure 3). Incorporating serial OI measurements into routine practice can improve early risk stratification, guide escalation of support, and inform decisions about rescue interventions. However, its role as a predictor of long-term morbidity, particularly neurocognitive outcomes, remains incompletely defined and warrants further investigation.


**Strengths and weaknesses of Berlin 2.0 and PALICC-2**


Both Berlin 2.0 and PALICC-2 preserve the one-week onset criterion, accept a broad range of underlying etiologies, and incorporate diagnostic pathways suitable for resource-limited settings. They also acknowledge the role of SpO_2_/FIO_2_ when arterial blood gases are unavailable, although they differ in the precise thresholds and in the clinical scenarios where this substitution is appropriate.

Berlin 2.0 stands out for its broad operational scope. By explicitly including patients on high-flow nasal oxygen and non-invasive ventilation within its severity classification, and by allowing the use of lung ultrasound as an imaging alternative, it greatly enhances applicability in diverse healthcare contexts. The definition also offers distinct diagnostic tracks for intubated, non-intubated, and low-resource settings, enabling flexibility in environments where advanced monitoring or imaging may not be available. This global implementability is further strengthened by the accommodation of oxygenation ratios and LUS when Arterial Blood Gas (ABG) or chest radiography are lacking, as well as by the removal of fixed PEEP requirements in resource-limited contexts. Importantly, Berlin 2.0 retains the PaO_2_/FIO_2_ metric and its established severity cutoffs, preserving continuity with decades of prior research and facilitating longitudinal comparisons.

However, these strengths come with some limitations. The requirement for bilateral opacities on imaging may inadvertently exclude clinically relevant cases with unilateral involvement [3], which would be considered under the pediatric criteria. While the inclusion of LUS expands diagnostic accessibility, it introduces operator-dependent variability [55], meaning that accuracy is strongly influenced by the skill and training of the clinician performing the scan. The reliance on SpO_2_/FIO_2_ ratios also carries the inherent biases and inaccuracies of pulse oximetry, particularly in patients with darker skin pigmentation, at high saturation levels (SpO_2_ > 97%), or in states of poor perfusion—circumstances in which ABG confirmation is advisable when important management decisions are at stake [45,46,47]. Furthermore, the prognostic value of Berlin 2.0’s severity classification in non-intubated patients remains uncertain and will require validation in prospective studies.

PALICC-2, in contrast, derives much of its strength from the physiological depth of its primary oxygenation metrics. Including mean airway pressure in OI and OSI captures both hypoxemia severity and the ventilatory burden needed to sustain it. This nuance makes OI and OSI more closely aligned with the pathophysiology of pediatric respiratory failure and more predictive of outcomes in invasively ventilated children. PALICC-2’s diagnostic structure further differentiates between “possible PARDS” in patients receiving high-flow or nasal interfaces and “full PARDS” in those on CPAP, BiPAP, or invasive ventilation, preventing the conflation of physiologically distinct groups. The explicit inclusion of patients with chronic pulmonary disease or cyanotic congenital heart disease, provided there is an acute deterioration from baseline, broadens clinical applicability and reflects the realities of pediatric intensive care. The acceptance of unilateral opacities in imaging is similarly pragmatic, acknowledging that focal patterns are common in children [56]. In adults the limitations of the imaging criteria are progressively being under debate. First the low sensitivity of chest radiographs [57] and the fact that an initial unilateral involvement can rapidly progress to bilateral infiltrates may miss an early diagnosis of ARDS. Second, severe respiratory failure with unilateral lung involvement shares many of the pathophysiologic, treatment and outcome characteristics of ARDS.

Nevertheless, PALICC-2 is not without challenges. The use of OI and OSI requires accurate, standardized measurement of mean airway pressure, and differences in ventilator management across centers may affect these values independent of the patient’s underlying lung injury. Like Berlin 2.0, PALICC-2’s reliance on SpO_2_ in OSI and SpO_2_/FIO_2_ carries the limitations of pulse oximetry [45,46,47], which can compromise accuracy in certain patient populations and clinical situations. The absence of lung ultrasound from its imaging criteria avoids the pitfalls of operator variability but removes a potentially valuable tool in contexts where chest radiography or CT are not feasible. Finally, the non-equivalence of severity thresholds and primary oxygenation metrics compared to Berlin 2.0 complicates direct cross-age epidemiological comparisons and the synthesis of data in multicenter or mixed-age research.

Therefore, Berlin 2.0 offers a globally adaptable framework that balances innovation with continuity, while PALICC-2 provides a physiologically targeted approach that may more accurately reflect disease severity in children. Both definitions make meaningful contributions toward aligning ARDS diagnosis with modern clinical practice, but each carries trade-offs between inclusivity, operational complexity, diagnostic precision, and comparability across age groups. Addressing these differences, especially in the context of collaborative research, remains an important challenge ahead.


**Gender considerations:**


While current ARDS definitions are gender-neutral, their implementation may perpetuate bias, particularly in ventilatory management. Systematic measurement of height, integration of predicted body weight calculators into clinical workflows, and tailored protocols for pregnancy could mitigate these disparities. Future research should prespecify sex-stratified analyses and actively address the evidence gap in pregnant patients to ensure equitable diagnosis, assessment, and management of ARDS.


**Key sources of heterogeneity and reporting standards**


Recently proposed consensus definitions have advanced a harmonized conceptual model of ARDS and broadened diagnostic inclusivity, but clinical adoption still requires prospective evidence of reliability and validity. Therefore, we propose a minimum validation framework comprising (i) standardized imaging reporting with inter-rater agreement; (ii) predefined HFNO settings; (iii) oxygenation metrics listed with well-defined ventilatory context (PEEP and MAP) and, when feasible, sensitivity analyses using OI/OSI, P/FP (i.e., PaO_2_/FIO_2_ adjusted for PEEP) [58], or mechanical power (MP)—an estimate of energy delivered by the ventilator per unit time [59].

To curb heterogeneity, we recommend structured chest radiograph/LUS interpretation (with double reading and competency statements), and an HFNO reporting set—device/interface, flow (L·min^−1^ and L·kg^−1^·min^−1^ in pediatrics), FIO_2_, cannula fit, patient position, sampling timing, and concurrent NIV/CPAP. Mechanical power has biological plausibility and prognostic validation, but methodologic nuances and pediatric data gaps argue for reporting it as a research metric rather than a diagnostic criterion at this stage. Thus, we believe that a transparent, standardized reporting will support reproducibility and the validation needed for safe implementation across adult and pediatric populations.

## 5. Conclusions

Current ARDS definitions for adults and children represent a substantial evolution towards frameworks that are more adaptable, physiologically informed, and globally applicable. Both aim to balance diagnostic precision with flexibility across a range of clinical contexts, yet their effective implementation will depend on local resources, training, and the capacity to collect standardized data. Future progress will hinge on harmonizing adult and pediatric criteria to facilitate cross-age research, validating prognostic tools like the oxygenation index in diverse populations, and integrating novel modalities—such as advanced imaging and non-invasive monitoring—without compromising feasibility in low-resource environments. Large-scale, prospective studies linking diagnostic criteria to both short- and long-term outcomes will be essential, not only to refine severity classification but also to ensure that definitions remain aligned with evolving evidence, technology, gender disparities and the overarching goal of improving survival and quality of life for patients with ARDS.

## Figures and Tables

**Figure 1 jcm-14-07644-f001:**
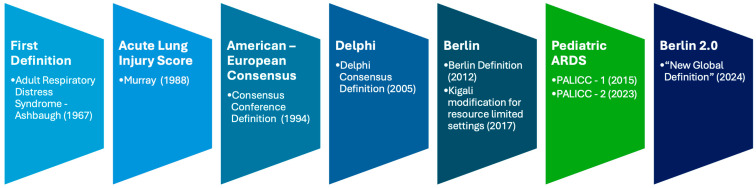
ARDS definitions timeline [1,2,5,6,7,8].

**Figure 2 jcm-14-07644-f002:**
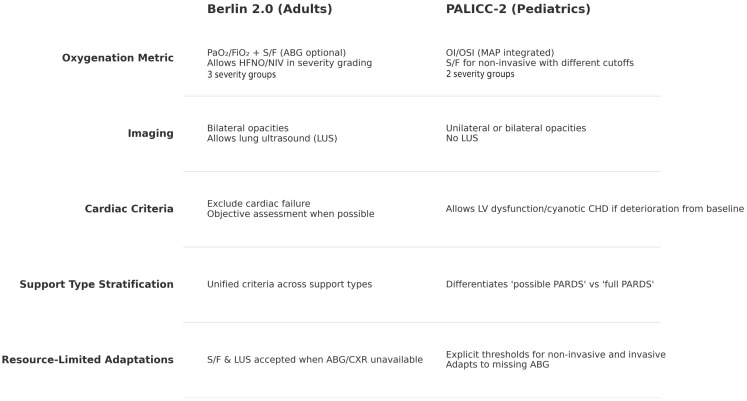
Main differences between Berlin 2.0 and PALICC-2 diagnostic criteria.

**Figure 3 jcm-14-07644-f003:**
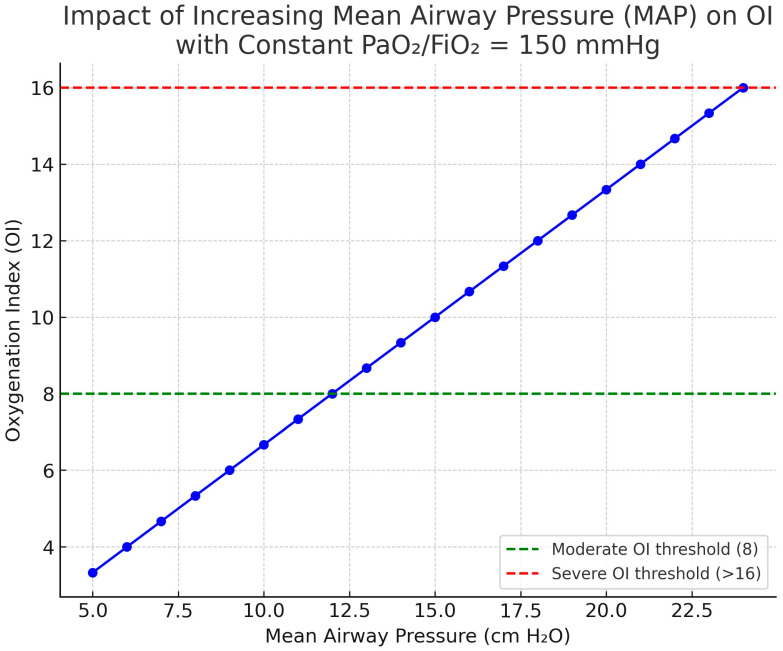
Influence of mean airway pressure (MAP) on the oxygenation index (OI). For a given PaO_2_/FIO_2_ (e.g., 150 mmHg for this example), MAP increments raise OI, reflecting the ventilatory burden required to sustain oxygenation. Thus, OI increases linearly together with MAP increments despite a constant PaO_2_/FIO_2_ of 150 mmHg. Abbreviations: MAP, mean airway pressure; OI, oxygenation index; OSI, oxygen saturation index; FiO_2_, fraction of inspired oxygen. OI = (MAP × FIO_2_ × 100)/PaO_2_; OSI is the saturation-based analogue using SpO_2_ instead of PaO_2_. Image generated using ChatGPT 5.0.

**Table 1 jcm-14-07644-t001:** Oxygenation criteria differences between Berlin 2.0 and PALICC-2 definitions.

Oxygenation Criteria	Berlin 2.0			PALICC-2	
**NIV**	**BiPAP/CPAP** **HFNO ≥ 30 L/min**			**Full Face +** **BiPAP/CPAP ≥ 5 cmH_2_O**	
				**Moderate**	**Severe**
	PaO_2_/FIO_2_ < 300 SpO_2_:FIO_2_ < 315			PaO_2_/FiO_2_ > 100 SpO_2_/FIO_2_ > 150	PaO_2_/FIO_2_ ≤ 100 SpO_2_/FIO_2_ ≤ 150
**Intubated Patients**	PaO_2_/FIO_2_ ≤ 300			OI ≥ 4 or OSI ≥ 5	
	**Mild**	**Moderate**	**Severe**	**Moderate**	**Severe**
PaO_2_/FIO_2_	>200–≤300	≤200–>100	≤100	OI < 16	OI ≥ 16
SpO_2_/FIO_2_	>235–≤315	≤235–>148	≤148	OSI < 12	OSI ≥ 12

## Data Availability

No new data were created or analyzed in this study.

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
