# Peer review of "Acute Respiratory Distress Syndrome Definitions in Adults and Children: A Comparative Narrative Review"

_jcm, 2025, doi:10.3390/jcm14217644_

Round 1
Reviewer 1 Report
Comments and Suggestions for Authors
Dear authors,
congratulations on this narrative review comparing the 2024 Berlin 2.0 (adult) and 2023 PALICC-2 (pediatric) definitions of ARDS.
The article effectively reviews the evolution of ARDS definitions, highlighting differences between adult (Berlin 2.0) and pediatric (PALICC-2) criteria across oxygenation thresholds, imaging, comorbidity inclusion, and applicability in resource-limited settings. It emphasizes the role of OI/OSI in pediatrics, bilateral vs. unilateral imaging requirements, and global implementation challenges. The authors conclude that harmonization is needed to improve research comparability and patient outcomes.
However, I have some major and minor comments:
Major comments
- The methods section briefly mentions a literature review, but more detail is warranted. Please expand on the literature search strategy (databases, keywords, inclusion/exclusion criteria, time frame) to strengthen reproducibility.
- Supplementary File 1 include only a list of acronyms and anything else. Should additional material be included in the supplementary file?
- Figure 3 (“Influence of Mean Airway Pressure (MAP) on Oxygenation Index (OI)”) requires more explanation. Please clarify its source and provide a more detailed description in the text.
Minor Comments:
- Ensure consistency in terminology (e.g., "SpOâ‚‚/FIOâ‚‚ ratio" vs. "SpO2:FIO2").
- Please check that all abbreviations are defined at first mention (though the list at the end is complete and helpful).
Some long sentences could be shorter, such as :
- Original: “In recent years, the growing role of non-invasive oxygenation indices, the increasing use of high-flow nasal oxygen (HFNO) especially during the COVID-19 pandemic, and calls to formally include HFNO in ARDS definitions, together with the expanding use of lung ultrasound in critical care, have challenged the boundaries of the Berlin criteria.”
Suggested: “Recently, the wider use of non-invasive oxygenation indices, HFNO (especially during COVID-19), and lung ultrasound has challenged the Berlin criteria and prompted calls to include these modalities in ARDS definitions.”
- Original: “By incorporating mean airway pressure into the oxygenation index and its saturation-based counterpart, the definition captures not only the degree of hypoxemia but also the ventilatory burden required to maintain it.”
Suggested: “Including mean airway pressure in OI and OSI captures both hypoxemia severity and the ventilatory burden needed to sustain it.”
Author Response
Reviewer 1
Thank you for your comments.
Major Comment 1 — Methods: literature search strategy needs more detail
Comment: The methods section briefly mentions a literature review, but more detail is warranted. Please expand on the literature search strategy (databases, keywords, inclusion/exclusion criteria, time frame) to strengthen reproducibility.
Response: We agree and have now expanded the Methods to provide a reproducible search strategy, including databases, full search strings, timeframe, inclusion/exclusion criteria, and screening/selection procedures. The full information is available in Supplemental Material 1 file.
Major Comment 2 — Supplementary File 1 content
Comment: Supplementary File 1 includes only a list of acronyms. Should additional material be included?
Response: Thank you for the suggestion. We have completely changed and expanded the Supplementary material as described previously.
Major Comment 3 — Figure 3 needs source and elaboration
Comment: Figure 3 requires more explanation. Please clarify its source and provide a more detailed description in the text.
Response: Thank you for raising the need for further explanation. Figure 3 illustrates how higher Mean Airway Pressure (MAP) increases Oxygenation Index (OI) for a given PaOâ‚‚/FiOâ‚‚ (eg. 150 mmHg in this example). We have added an extended comment and legends for a better understanding.
Minor Comment A — Terminology consistency
“Ensure consistency in terminology (e.g., "SpOâ‚‚/FIOâ‚‚ ratio" vs. "SpO2:FIO2").”
Response: Agreed. We standardized to “SpOâ‚‚/FIOâ‚‚ ratio” and harmonized capitalization along the text.
Minor Comment B — Define abbreviations at first mention
“Please check that all abbreviations are defined at first mention (though the list at the end is complete and helpful”
Response: Done. We ensured first-mention definitions for all abbreviations. We have expanded the abbreviation list at the end of the manuscript.
Minor Comment C — Shorten long sentences (two examples)
Response: Thank you for your suggestions, which we adopted and implemented in the text for clarity.

Reviewer 2 Report
Comments and Suggestions for Authors
Thank you for this narrative review. Although the idea of harmonising definitions is excellent, the manuscript lacks depth. The challenges of the current definitions are discussed in the recent international Delphi study (doi: 10.1016/S2213-2600(25)00115-8.) There are many limitations with the current definition and the process by which the definition was drafted. The reliability and validity of the new Global definition still need to be proved, and it is a significant step before this can be adopted in clinical settings. Without acknowledging the gaps, such as the effect of ventilation on PaO2/FiO2 and SPO2/FiO2, inter-individual variations in the interpretation of chest X-rays and Lung ultrasound, and settings of HFNC for the diagnosis of ARDS, the heterogeneity will only widen and may limit the research.
Author Response
Reviewer 2
Thank you for your comments. Therefore, we have drafted a completely new subsection in the discussion of our manuscript addressing the main three concerns of the reviewer.
"Thank you for this narrative review. Although the idea of harmonising definitions is excellent, the manuscript lacks depth. The challenges of the current definitions are discussed in the recent international Delphi study (doi: 10.1016/S2213-2600(25)00115-8.) There are many limitations with the current definition and the process by which the definition was drafted. The reliability and validity of the new Global definition still need to be proved, and it is a significant step before this can be adopted in clinical settings. Without acknowledging the gaps, such as the effect of ventilation on PaO2/FiO2 and SPO2/FiO2, inter-individual variations in the interpretation of chest X-rays and Lung ultrasound, and settings of HFNC for the diagnosis of ARDS, the heterogeneity will only widen and may limit the research."
Major Comment 1 — Depth and critical appraisal of current/global definitions
Response: we thank the reviewer for this comment. An in depth review and assessment of the challenges and limitations of the current definitions was not the primary scope of our manuscript. We fully agree with the existing limitations and the potential for the necessary future improvements of the definition such as standardizing oxygenation assessment (which we mentioned in the manuscript) ideally integrating oxygenation criteria with other relevant variables such as compliance, lung efficiency (CO2 derived indices) or sub-phenotyping which previous consensus definitions have not been able to or failed to consider primarily because of the methodology used to set the definiton.
To address your request, we have now expanded the Discussion with a subsection appraising recent consensus disparities and harmonization potential flaws, and outlined a minimum validation framework.
Major Comment 2 — Unaddressed gaps that risk widening heterogeneity
Response: We expanded the discussion section describing key sources of heterogeneity with concrete reporting standards (ventilatory context for PaO2/FiO2 and SpO2/FiO2; imaging reliability for Chest X-Ray and LUS; HFNO minimum reporting set; adult–pediatric divergences with OI/OSI), and added two additional ventilatory standards to reflect the burden of the mechanical ventilation.
Tone and scope of the manuscript
Response: To address this concern, we emphasize the need to balance harmonization with diagnostic specificity, and added a brief research agenda.

Round 2
Reviewer 1 Report
Comments and Suggestions for Authors
I have no further comments
Author Response
Thank you
Reviewer 2 Report
Comments and Suggestions for Authors
Thank you for the revisions and responses to the concerns raised. However, this doesn't address the major challenges with the definitions of ARDS and the steps ahead. There is no new message for the readers.
Author Response
Response to Reviewer 2
Thank you for your follow-up. We share your concern that the field needs clearer direction on how current ARDS definitions should evolve and be operationalized, as described in the Discussion section "Strengths and weaknesses of Berlin 2.0 and PALICC-2". Moreover, in this revision we aimed to make three concrete contributions:
-
A minimum validation framework: standardized imaging adjudication with agreement metrics; predefined HFNO settings/reporting; oxygenation metrics paired with ventilatory context (PEEP/MAP) and sensitivity analyses (OI/OSI, P/FP); and adult–pediatric stratified validation across settings.
-
A core reporting set to curb heterogeneity (Discussion, “Key sources of heterogeneity and reporting standards”: what to report for HFNO, imaging reliability, timing of sampling, and parallel oxygenation metrics.
-
A focused research agenda that outlines immediate next steps for prospective validation and threshold optimization in adults and children.
If there are additional specific gaps you would like us to address, we would be grateful for your guidance and are ready to incorporate further revisions.